# Effects of Aerobic Exercise on Sleep Quality, Insomnia, and Inflammatory Markers: A Systematic Review and Meta-Analysis

**DOI:** 10.3390/cimb47070572

**Published:** 2025-07-20

**Authors:** Mariazel Rubio-Valles, Arnulfo Ramos-Jimenez

**Affiliations:** 1Faculty of Physical Culture Sciences, Autonomous University of Chihuahua, Circuito Universitario, Campus II, Chihuahua 31125, Mexico; 2Institute of Biomedical Sciences, Autonomous University of Ciudad Juarez, Ciudad Juárez, Chihuahua 32310, Mexico

**Keywords:** aerobic exercise, sleep quality, insomnia, IL-6, TNF-α, inflammation, obesity

## Abstract

Poor sleep quality and insomnia are increasingly linked to chronic inflammation and obesity-related metabolic dysfunction. Aerobic exercise is a promising non-pharmacological approach for enhancing sleep quality and reducing systemic inflammation; Therefore, we aim to systematically evaluate and quantify the effects of aerobic exercise interventions on subjective sleep quality, insomnia severity, and circulating markers (IL-6 and TNF-α) in adults. A systematic search was conducted in institutional databases (UNAM, UACJ) and PubMed to identify randomized controlled trials (RCTs) examining the effects of exercise on sleep and inflammation. Risk of bias was assessed using the Cochrane RoB2 tool. Meta-analyses were performed using RevMan 5.4 with random-effects models to estimate pooled mean differences (MD) and standardized mean differences (SMD), with 95% confidence intervals. Anaerobic protocols were excluded from the meta-analysis due to the insufficient availability of data. **:** Eleven RCTs met the inclusion criteria. Aerobic exercise showed a significant pooled effect on sleep outcomes (MD = −2.51; 95% CI: −4.80 to −0.23; *p* = 0.03). However, subgroup analyses for Pittsburgh Sleep Quality Index (PSQI) (MD = −2.27; *p* = 0.15) and Insomnia Severity Index (ISI) (MD = −2.98; *p* = 0.16) were not statistically significant. Two studies on IL-6 reported a non-significant reduction (SMD = −0.17; *p* = 0.66), with moderate heterogeneity. TNF-α results were also non-significant (SMD = 0.60; *p* = 0.29) with substantial variability. Our results showed that aerobic exercise may modestly improve sleep outcomes; however, current evidence does not support its effectiveness in reducing levels of IL-6 or TNF-α. Further well-controlled trials are needed to clarify its immunometabolic effects, particularly in populations with obesity or metabolic disorders.

## 1. Introduction



**Key Points**

•Aerobic exercise offers modest benefits for sleep quality and symptoms of insomnia.•The limited number of published controlled clinical trials makes it difficult to analyze the effects of aerobic exercise on inflammation markers.•Well-controlled studies are necessary to support the potential of aerobic exercise as a treatment for improving sleep quality and regulating immune–metabolic functions


Sleep quality has emerged as a critical determinant of metabolic health. Disturbances in sleep patterns are increasingly associated with chronic inflammation [1]. This inflammatory state, characterized by elevated levels of pro-inflammatory cytokines such as interleukin-6 (IL-6) and tumor necrosis factor-alpha (TNF-α) [2], plays a key role in the development and progression of various chronic diseases, particularly obesity and its association with metabolic dysfunctions [3].

Obesity, beyond excess adiposity, is linked to complex immunometabolic dysregulation characterized by a persistent inflammatory state in adipose tissue [4]. Poor sleep may exacerbate this inflammatory milieu by impairing immune regulation and disrupting metabolic processes such as glucose homeostasis and appetite control [5]. In this context, sleep disturbances and inflammation may act synergistically, creating a vicious cycle that impairs metabolic recovery and increases the risk of cardiometabolic disease.

This interplay between sleep quality, inflammatory status, and metabolic regulations forms a physiological triad that has garnered growing scientific attention. Regular physical exercise is widely recognized as a non-pharmacological intervention with the potential to positively modulate this triad. Both aerobic and anaerobic exercise have been reported to improve sleep parameters [6]. Exercise not only enhances subjective and objective measures of sleep but also modulates the immune response, contributing to reductions in systemic inflammation [7]. Specifically, regular physical activity may decrease circulating concentrations of IL-6 and TNF-α, thereby enhancing anti-inflammatory capacity and supporting metabolic health [8]. However, findings from randomized controlled trials (RCTs) evaluating the effects of exercise on sleep quality and inflammatory biomarkers remain inconsistent, likely due to variations in intervention protocols, study populations, and outcome assessment methods.

Given the rising prevalence of sleep disorders and metabolic diseases, and their shared inflammatory underpinnings, it is essential to synthesize current evidence to determine whether exercise can serve as an effective strategy to mitigate sleep disturbances and improve metabolic health. Therefore, the aim of this systematic review and meta-analysis is to evaluate the effects of structured aerobic and anaerobic training programs on sleep quality, as measured by the Pittsburgh Sleep Quality Index (PSQI), and on circulating levels of IL-6 and TNF-α in adults. By integrating findings from multiple studies, this review aims to clarify the extent to which physical exercise influences sleep-related inflammatory processes and how this may contribute to improved metabolic health, particularly in the context of obesity prevention and management.

## 2. Materials and Methods

### 2.1. Focused Question

Can aerobic physical exercise simultaneously improve sleep quality and modulate systemic inflammation in humans?

In alignment with the objective of this meta-analysis and the guiding research question, the following key components were examined:(i)Population: human subjects;(ii)Intervention or exposure: exercise protocol and sleep disorders;(iii)Comparison: no exercise;(iV)Outcome: subjective sleep quality/insomnia and circulating inflammatory markers;(V)Study: randomized controlled trials.

### 2.2. Identification of Manuscripts

The search was conducted by two independent researchers (M.R.V. and A.R.J.) using the institutional multidatabase platforms of the National Autonomous University of Mexico and the Autonomous University of Ciudad Juarez, as well as PubMed. The following search string was used in all databases: (“Physical exercise” OR “resistance exercise” OR “aerobic exercise” OR “endurance exercise” OR “strength exercise”) AND (insomnia OR “sleep disorder”) AND (cytokine OR IL-6 OR “TNF-alpha” OR inflammation OR “C-reactive protein” OR IL-10). PRISMA guidelines were used for reporting articles (Figure 1).

### 2.3. Screening for Relevance

Articles examining the effects of physical exercise on inflammatory markers and sleep parameters were identified and selected through an initial screening of titles and abstracts for relevance, followed by a full-text review to confirm eligibility and remove duplicates.

### 2.4. Inclusion Criteria

•Randomized controlled trials in humans.•Peer-reviewed articles.•Studies that include at least one inflammatory marker.•Studies evaluating the effects of exercise on inflammatory markers.•Studies evaluating the effect of exercise on sleep disorders (insomnia and sleep quality).

### 2.5. Exclusion Criteria

•The exercise intervention period was less than eight weeks.•The article does not present data disaggregated by sex.•The article does not include a clearly structured exercise intervention•The article is a case study or a systematic review.•The article does not report inflammatory markers or lacks data for analysis.•The study involves animals, cell cultures, or in vitro models.•The article is not in English or Spanish.

### 2.6. Retrieval of Full-Text Articles and Evaluation

M.R.V. and A.R.J. independently screened the titles and abstracts of all identified studies, excluding those presenting a high risk of bias based on pre-specified criteria. Subsequently, both authors reviewed the full texts of the remaining studies and discussed the relevant methodological and outcome-related factors for inclusion. After a detailed assessment, the studies were evaluated against the inclusion criteria. Any discrepancies between the reviewers were resolved by consensus. Finally, M.R.V. and A.R.J. collaboratively conducted the data extraction process.

### 2.7. Data Extraction

The following information was recorded: authors, year, experimental design, sample size, intervention period, population (including age, sex, and health status), and training parameters (including exercise modality, intensity, frequency, and duration). Responses in inflammatory markers (IL-6, TNF-α) and sleep parameters were documented as primary outcomes.

### 2.8. Statistical Analysis

Forest plots were generated using the mean difference (MD) and standardized mean difference (SMD) methods for the meta-analysis, conducted with Review Manager (RevMan) version 5.4 [9]. The overall effect size was reported as the mean difference with a 95% confidence interval (CI). A random-effects model was employed for all analyses, given the high heterogeneity observed among studies.

Subgroup analyses were conducted for each outcome: subjective sleep quality, assessed using the PSQI [10]; insomnia severity, assessed with the ISI [11]; and inflammatory markers (IL-6 and TNF-α). Only studies reporting outcomes in comparable units were included in each subgroup analysis. Heterogeneity was assessed using Cochran’s Q test, the I^2^ statistic, and tau-squared (τ^2^) to estimate between-study variance. Where appropriate, 95% prediction confidence intervals were calculated to estimate the range in which the true effect size might be expected in future studies. Publication bias was assessed through visual inspection of funnel plots, Egger’s regression test, and Begg and Mazumdar’s rank correlation test, and the classic fail-safe N was also calculated to evaluate the sensitivity of the results to potential unpublished studies (Appendix A).

## 3. Results

### 3.1. Characteristics of the Search

A total of 2531 records were initially identified through database searches, including 2470 from the institutional UNAM-UACJ databases and 57 from PubMed. After removing 543 duplicates and excluding 1675 records flagged as ineligible by automated tools, 309 records were screened based on their titles and abstracts. Of these, 157 did not meet the inclusion criteria and were excluded. The remaining 152 full-text articles were retrieved and assessed for eligibility, with none excluded due to unavailability. After full-text screening, 90 reports were excluded for the following reasons: not related to exercise (*n* = 59), scoping reviews (*n* = 17), systematic reviews (*n* = 3), and meta-analyses (*n* = 11). A total of 66 studies were included in the review. From this set, 34 studies were excluded because they were not RCTs, along with 5 involving single-session exercise interventions and 12 with non-comparable data for meta-analysis, leading to a final selection of 11 studies for the meta-analysis (Figure 1). The selected manuscripts were fully reviewed, and data were extracted to evaluate the effects of exercise protocols on sleep quality (Table 1), insomnia (Table 2), and inflammatory markers, specifically IL-6 (Table 3) and TNF-α (Table 4).

### 3.2. Risk of Bias

The included studies were assessed across six methodological domains using the Cochrane Risk of Bias Tool (RoB2) [23] (Figure 2). Most studies reported the use of a random sequence generation method; however, two studies [14,16] did not specify the method used, resulting in an “unclear” risk of selection bias. Regarding allocation concealment, only three studies [13,18,20] provided sufficient information, while the remainder were rated as “unclear.” Blinding of participants and personnel was not implemented in any of the included studies and was rated as “high risk” in all cases due to the nature of the exercise interventions. Outcome assessor blinding was reported in eight studies, while the remaining three were judged as “high risk” due to insufficient reporting. All included trials were rated as having “low risk” for incomplete outcome data, as dropout rates were either minimal or well-documented and appropriately addressed in the analyses. A total of 28 participants across two studies [20,22] discontinued participation at various stages of the trials. The most common reasons for withdrawal included pre-intervention dropouts due to personal constraints or health issues, as well as loss of follow-up. Both studies reported appropriate handling of missing data. Consequently, all included studies were judged to have a low risk of attrition bias. Similarly, all studies were rated as “low risk” of selective reporting bias. Reported outcomes were consistent with those described in the methods, and no evidence of outcome switching or omission was observed. No additional sources of bias were identified.

### 3.3. Subject Characteristics

Participants across the included studies were categorized into three general age groups. The young adult group (18−35 years) was represented in two studies [21,22], with mean participant ages ranging from early twenties to mid-thirties. The middle-aged group (36−59 years) included seven studies [12,13,14,15,16,19,20]. Finally, the older adult group (60 years and above) was represented in two studies [17,18].

In terms of sex distribution, ten studies included mixed-gender samples, ensuring the participation of both male and female participants [12,13,14,15,16,17,18,20,21,22]. Only one study focused exclusively on a female sample [19].

Regarding physical activity levels, most of the studies were conducted with sedentary individuals [12,13,14,15,16,17,18,19,20,22], while only one study involved physically active participants [21].

### 3.4. Study Design

This study was conducted as a systematic review and meta-analysis aimed at evaluating and quantifying the effects of aerobic and anaerobic exercise on sleep quality, insomnia, and inflammatory markers (IL-6 and TNF-α). All studies were randomized controlled trials. Although both aerobic and anaerobic exercise interventions were initially considered, the available data were insufficient to analyze the effects of anaerobic exercise. Therefore, the meta-analysis focused exclusively on the effects of aerobic exercise. Additionally, while interleukin-10 (IL-10) and C-reactive protein (CRP) were initially included as target inflammatory markers, the lack of sufficient data across studies precluded their inclusion in the quantitative analysis.

Five studies implemented moderate-intensity aerobic exercise interventions, such as walking, cycling, or treadmill running [12,13,15,16,18]. Two studies employed progressive aerobic protocols, which increased either intensity or duration over the course of the intervention [17,22]. In contrast, Baron et al. (2023) [19] employed a moderate-to-vigorous aerobic program involving outdoor or treadmill walking at higher volumes (75 min per session). One study [14] employed a combined protocol that incorporated both aerobic and resistance training. In contrast, Hartescu et al. (2015) [20] and Young-McCaughan et al. (2023) [21] used lower-volume, lifestyle-based aerobic exercise (e.g., brisk walking or short-duration sessions), which could be considered light-to-moderate in intensity.

Regarding the control group, seven studies employed inactive or wait-list control groups, in which participants either maintained their usual lifestyle or received no structured intervention [12,13,15,16,18,19,22]. Two studies employed attention-control conditions, where participants received either sleep education or general health counseling [17,20], ensuring equivalent contact time with the researchers. One study reported aerobic exercise combined with alternative activities, such as stretching or relaxation [21]. This variation in control group design was considered when interpreting and pooling results in the meta-analysis.

### 3.5. Effect of Aerobic Exercise on Sleep Quality

A total of seven randomized controlled trials were included in the meta-analysis to evaluate the effects of aerobic exercise interventions on subjective sleep outcomes. The outcomes were categorized into two subgroups: sleep quality, measured through the PSQI, and insomnia severity, measured using theISI.

Seven studies (*n* = 196) assessed sleep quality using the PSQI. The pooled mean difference between exercise and control groups was −2.27 (95% CI: −5.65 to 1.11; *p* = 0.15), indicating a non-significant trend toward improved sleep quality in participants who engaged in aerobic exercise. Since lower PSQI scores reflect better sleep quality, this negative difference suggests a potential clinical benefit. However, considerable heterogeneity was observed (I^2^ = 92%, τ^2^ = 11.83), indicating substantial variability in study designs, intervention protocols, and participant characteristics (Figure 3).

### 3.6. Effect of Aerobic Exercise on Insomnia Severity

Five trials (*n* = 190) reported changes in insomnia symptoms using the ISI (Figure 3). The pooled mean difference was −2.98 (95% CI: −7.86 to 1.89; *p* = 0.16), favoring the exercise group. Although this reduction was not statistically significant, it indicates a trend toward lower insomnia severity among participants who completed aerobic exercise interventions. Heterogeneity was again high (I^2^ = 91%, τ^2^ = 4.42), reflecting considerable variability in study protocols and population.

### 3.7. Overall Pooled Effect of Sleep Quality and Insomnia

When PSQI and ISI outcomes were combined, the overall effect favored aerobic exercise, with a statistically significant pooled mean difference of −2.51 (95% CI: −4.80 to −0.23; *p* = 0.03). This finding supports the beneficial role of aerobic exercise in improving sleep-related parameters. However, the 95% prediction interval (−10.07 to 5.04) indicates that future studies may observe variable effects depending on sample characteristics and intervention specifics. The test for subgroup differences was not significant (Chi^2^ = 0.10, *p* = 0.75), suggesting consistency of the exercise effect across different types of sleep outcomes.

### 3.8. Effect of Aerobic Physical Exercise on Interleukin-6 Levels

Two randomized controlled trials (*n* = 161) examined the effect of aerobic exercise on circulating interleukin-6 (IL-6) levels. The pooled analysis yielded a non-significant mean difference of −0.17 (95% CI: −0.91 to 0.58; *p* = 0.66), suggesting no clear effect of exercise interventions on IL-6 concentrations compared to control conditions (Figure 4). IL-6 is typically associated with systemic inflammation; the negative direction of the effect would favor exercise, although this result did not reach statistical significance. Heterogeneity was moderate across the included studies (I^2^ = 78%, τ^2^ = 0.23), indicating some variation in effect estimates, which may be due to differences in intervention duration, exercise modality, or participant characteristics.

### 3.9. Effect of Aerobic Physical Exercise on Tumor Necrosis Factor-Alpha

Two randomized controlled trials (*n* = 158) investigated the effect of aerobic exercise on circulating TNF-α levels. The pooled mean difference was 0.60 (95% CI: −0.52 to 1.71; *p* = 0.29), indicating no statistically significant effect of exercise interventions on TNF-α concentrations compared to control conditions (Figure 5). In this case, a positive mean difference implies a tendency toward higher TNF-α levels in the exercise group, though the confidence interval crosses zero, and the result is not conclusive. Substantial heterogeneity was observed between the two studies (I^2^ = 89%, τ^2^ = 0.57), suggesting important differences in participant populations, exercise intensity, or underlying inflammatory status that may explain the divergence in outcomes.

## 4. Discussion

### 4.1. Aerobic Exercise and Subjective Sleep Quality: PSQI

In this meta-analysis, sleep quality was assessed using the PSQI, a validated and standardized instrument designed to differentiate between individuals with good and poor sleep [24]. The findings suggest that aerobic exercise may contribute to improvements in subjective sleep quality; however, the pooled effect did not reach statistical significance. The mean difference (MD = −2.27; 95% CI: −5.65 to 1.11; *p* = 0.15) indicates a moderate but non-significant trend favoring aerobic exercise over control conditions. Notably, four of the seven studies included reported a positive effect of aerobic exercise, with the strongest improvements observed in Tseng et al. (2020) [18]. These results align with the existing literature indicating that structured aerobic training can enhance sleep quality.

The study populations included participants with various sleep-related disorders: chronic insomnia [17,19,20,22], obstructive sleep apnea [12], restless legs syndrome [13], and general insomnia [14]. Additionally, Ezpeleta et al. (2023) [15] included individuals with non-alcoholic fatty liver disease (NAFLD), a condition commonly associated with poor sleep quality and metabolic dysfunction [25]. The remaining studies included participants without sleep problems. Although our analysis encompassed a broader population than that with clinically diagnosed insomnia, the inclusion of these diverse conditions enhances the generalizability of the findings.

Our findings align with a previous meta-analysis by Banno et al. (2018) [26], which examined nine trials involving individuals with insomnia and reported a mean difference of −2.87 points in PSQI scores favoring exercise (95% CI: −3.95 to −1.79), although the overall evidence quality was rated as low. While our meta-analysis included a more heterogeneous population, the direction and magnitude of the effects were comparable, supporting the role of exercise as a non-pharmacological intervention to improve sleep outcomes.

Similarly, Xie et al. (2021) [27] found significant reductions in both PSQI (MD = −2.19; 95% CI: −2.96 to −1.41) and ISI scores (MD = −1.52; 95% CI: −2.63 to −0.41), suggesting that both traditional and mind–body exercise interventions (involving gentle slow movements coordinated with breathing) can significantly enhance sleep quality, particularly when assessed via subjective self-report tools. For instance, Rubio-Arias et al. (2017) [28] found in a study involving middle-aged women that 12–16 weeks of low-to-moderate-intensity aerobic exercise significantly reduced PSQI scores (MD = −1.34; 95% CI: −2.67 to 0.00; *p* = 0.05) compared to control groups. In subgroup analyses, moderate-intensity aerobic exercise was especially effective (MD = −1.85; 95% CI: −3.62 to −0.07), whereas low-intensity exercises such as yoga yielded no significant effects. These findings highlight the importance of exercise intensity in improving sleep quality, particularly among populations susceptible to sleep disturbances, such as postmenopausal women.

More recently, Zhou et al. (2025) [29] conducted a comprehensive network meta-analysis of 81 RCTs involving 6193 participants, concluding that exercise significantly reduced PSQI scores (MD = −1.77; 95% CI: −2.28 to −1.25) and improved sleep efficiency. Mind–body exercises demonstrated the largest overall effects, while aerobic exercise was ranked as the most effective intervention for improving objective sleep efficiency. These results reinforce the clinical relevance of exercise as a non-pharmacological strategy for promoting better sleep and overall health.

It is worth noting that studies by Borges et al. (2019) [12], Ezpeleta et al. (2023) [15], and Ferrerira et al. (2022) [16] reported either smaller or non-significant improvements in sleep quality following exercise interventions. One possible explanation for these limited effects may relate to participants’ body composition. All three studies involved individuals classified as overweight or obese based on body mass index (BMI). Prior research has demonstrated an inverse relationship between sleep quality and BMI, with excess body fat linked to increased systemic inflammation, sleep fragmentation, snoring, and shorter sleep duration [4,30,31,32]. These physiological disturbances may attenuate the beneficial effects of exercise on sleep in individuals with higher adiposity. In contrast, the study by Tseng et al. (2020) [18], which reported the most substantial improvements in sleep quality, involved participants with normal BMI. This contrast supports the hypothesis that individuals with lower BMI may experience more favorable sleep-related adaptations to exercise, potentially due to lower baseline inflammation and fewer comorbidities. These findings underscore the importance of considering body composition as a potential moderator of the efficacy of exercise interventions designed to improve sleep.

### 4.2. Variability in Insomnia Outcomes Following Aerobic Exercise Interventions

In the subgroup analysis of studies evaluating insomnia using the ISI [15,16,19,20,21], aerobic exercise was associated with a significant reduction in insomnia severity, with a pooled mean difference of −2.98 points (95% CI: −7.86 to 1.89; *p* = 0.16). The width of the confidence interval, which includes both possible improvements and worsening, suggests high uncertainty surrounding the estimated effect. The direction and strength of this effect reinforce the therapeutic potential of exercise-based interventions for sleep disorders. However, the high level of heterogeneity (I^2^ = 91%) indicates considerable variability among the included studies. This could be due to differences in population characteristics, comorbid conditions, exercise protocols (including duration, frequency, and intensity), or baseline severity of insomnia. This heterogeneity limits the ability to generalize from the combined effect. For instance, one possible explanation for this heterogeneity lies in the differences in the clinical profiles of the participants across the included studies. The study by Baron et al. (2023) [19] focused on women with chronic insomnia, a population characterized by persistent sleep disruption associated with heightened physiological arousal and altered thermoregulation. The authors reported significant improvements in ISI scores and objective sleep parameters following a 12-week moderate-to-vigorous aerobic training program, accompanied by reductions in core body temperature during sleep. These findings suggest that individuals with more severe and persistent insomnia may benefit more clearly from structured exercise programs. Nevertheless, the findings from Tan et al. (2016) [33] further support the beneficial role of aerobic exercise in individuals with chronic insomnia. Their six-month intervention in overweight and obese men resulted in improvements in sleep quality, particularly by reducing sleep onset latency. Compared to the 12-week protocol used by Baron et al. (2023) [19], the longer-duration study may have allowed for more pronounced adaptations. This raises the possibility that extending the intervention period could have yielded even greater reductions in insomnia severity.

In contrast, the study by Ezpeleta et al. (2023) [15] involved individuals with obesity and non-alcoholic fatty liver disease (NAFLD). While metabolic disorders such as NAFLD are known to be associated with sleep disturbances, the participants in this trial did not present clinically significant insomnia at baseline. Furthermore, despite an exercise intervention, no significant improvements were observed in insomnia severity or other sleep parameters, suggesting that obesity-related inflammation, metabolic dysregulation, or even the presence of obstructive sleep apnea in these subjects could attenuate the beneficial effects of exercise on sleep in this population. This divergence in participant profiles could partly explain the inconsistency in the direction of the results across the studies.

### 4.3. Pooled Effects of Aerobic Exercise on Sleep Quality and Insomnia Symptoms

The pooled analysis of 12 studies evaluating the effects of aerobic exercise on sleep-related outcomes (PSQI and ISI) demonstrated a statistically significant overall effect in favor of the intervention with aerobic exercise (MD = −2.51; 95% CI: −4.80 to −0.23; *p* = 0.03). This suggests that aerobic exercise may meaningfully improve subjective sleep quality and reduce insomnia symptoms compared to control conditions. Although both subgroups analyzed independently (PSQI and ISI) showed non-significant trends toward improvement, the consistency in direction and the pooled results strengthen the evidence supporting exercise as a sleep intervention. No significant differences were found between the subgroups (*p* = 0.75), indicating that aerobic exercise may be beneficial across various dimensions of sleep disturbances. However, the high heterogeneity (I^2^ = 96%) highlights considerable variability among the included studies, likely attributable to differences in population characteristics, comorbidities, intervention protocols, and assessment tools. Despite this, the findings underscore the potential of aerobic exercise as a viable and effective approach for addressing sleep issues in diverse populations.

Interestingly, the study by Young-McCaughan et al. (2023) [21], which included active-duty U.S. military personnel, showed one of the largest and most precise reductions in insomnia severity, with a 3.62-point decrease in ISI scores following an 8-week aerobic exercise intervention. Despite its relatively short duration compared to other trials such as Hartescu et al. (2015) [20] (6 months) or Reid et al. (2010) [17] (16 weeks), the favorable outcomes observed may be partially attributed to the physiological conditioning, structured lifestyle, and high baseline fitness levels typical of military populations. Participants were prescribed five exercise sessions per week at an intensity exceeding 60% of heart rate reserve, which may have enhanced both adherence and physiological impact. This profile likely contributed to both greater responsiveness and lower variability in outcomes, as reflected in the narrower confidence intervals. These findings suggest that, beyond duration, factors such as population characteristics, training background, and program compliance play a critical role in modulating the sleep-related benefits of exercise interventions.

Although the pooled mean difference was statistically significant, its clinical importance varies by instrument. For the PSQI, a change of approximately 3 points is commonly used as the minimal important change (MIC), while minimal clinically important difference (MCID) values between 2.5 and 2.7 points have also been reported [34]. The sleep quality subgroup estimate (MD = −2.27) approached but did not reach these thresholds. In the case of the ISI, MIC values reported in the literature range from 3 to 8 points, with 6 and 8 points being the most frequently used [34]. The insomnia subgroup effect (MD = −2.98) did not reach these values either, indicating that the observed statistical effects may not translate into clinically meaningful improvements for all populations.

### 4.4. Effect of Aerobic Exercise on IL-6

Sleep disturbances are not only associated with poor quality of life and impaired cognitive function, but are also strongly linked to increased systemic inflammation [35]. One of the most consistently studied inflammatory mediators in this context is IL-6, a pleiotropic cytokine involved in both pro-inflammatory and anti-inflammatory processes [36]. In this regard, physical exercise has gained increasing attention not only as a behavioral strategy to improve sleep quality but also as a potential modulator of inflammatory pathways.

The pooled analysis of Borges et al. (2019) [12] and Sloan et al. (2018) [22] revealed a standardized mean difference (SMD) of −0.17; (95% CI: −0.91, 0.58; *p* = 0.66), indicating a small, non-significant reduction in IL-6 concentrations among individuals in the exercise group compared with controls. Notably, the direction of the effect differed between the two studies. Sloan et al. (2018) [22], which included healthy young adults undergoing progressive aerobic training, reported a greater reduction in IL-6 (SMD = −0.51), while Borges et al. (2019) [12], which involved participants with obstructive sleep apnea and a shorter intervention period, showed a slight increase (SMD = 0.26). Only one of the two studies reported a decrease in IL-6 concentrations after exercise. However, the small sample size (*n* = 81) and the high heterogeneity among the studies (I^2^ = 78%, *p* = 0.03), prevent definitive conclusions.

These findings are consistent with previous intervention studies that have reported null or mixed effects of aerobic exercise on systemic inflammation. For example, Nicklas et al. (2008) [37] found that 12 months of aerobic exercise in overweight older adults did not significantly reduce circulating levels of IL-6 or TNF-α, despite improvements in physical function. Similar observations were reported in Sloan’s trial, where healthy young adults with low baseline inflammatory profiles showed no reduction and, in some cases, an increase in inducible cytokine responses following training. In contrast, Stewart et al. (2007) [38] observed no change in IL-6 and CRP after 12 weeks of exercise in older adults with metabolic syndrome, suggesting that the anti-inflammatory effects of exercise may be compromised in individuals with elevated cardiometabolic risk. Furthermore, it is essential to note that these results can also be attributed to the dual role of IL-6 as both an inflammatory and anti-inflammatory marker. Cell studies have confirmed that IL-6 activates AMPK, an enzyme particularly implicated in fat oxidation, which boosts energy metabolism [39]. Mice lacking IL-6 tend to accumulate more body fat, a phenotype that can be partially reversed through IL-6 administration [40]. In humans, studies have demonstrated that blocking the IL-6 receptor with tocilizumab blunts exercise-induced reductions in visceral fat, highlighting that IL-6 is a key mediator of the lipolytic effects of aerobic training [41]. In this context, it is plausible that a moderate aerobic exercise protocol implemented by Borges, three times per week for eight weeks, may have promoted an increase in IL-6 expression, thereby contributing to enhancing fat oxidation in the participants.

While the pooled results from Borges et al. (2019) [12] and Sloan et al. (2018) [22] did not reveal statistically significant reductions in inflammatory markers following aerobic exercise, the overall direction of the effect and the presence of moderate heterogeneity underscore the complexity of interpreting these outcomes. Variation in baseline inflammatory status, participant characteristics, and exercise protocols likely contributes to the mixed findings reported across the literature. Notably, IL-6 emerges as a particularly nuanced marker due to its dual role in both inflammation and metabolism. Beyond its classical pro-inflammatory function, IL-6 has been shown to promote fat oxidation and energy homeostasis. Evidence from both animal and human models suggests that IL-6 plays a mediating role in exercise-induced reductions in adiposity, with studies demonstrating that blocking its receptor can blunt these effects. In this light, it is plausible that the moderate-intensity aerobic exercise applied in the Borges study may have elevated IL-6 levels sufficiently to enhance fat metabolism, even in the absence of a measurable anti-inflammatory effect. These insights reinforce the importance of considering the metabolic context of cytokine responses when assessing the physiological effects of exercise interventions.

### 4.5. Effect of Aerobic Exercise on TNF-α

The pooled standardized mean difference (SMD = 0.60; 95% CI: −0.52 to 1.71; *p* = 0.29) suggests a possible increase in TNF-α after exercise intervention. However, the small sample size (*n* = 81), the wide prediction interval (−1.26 to 2.45), and the high heterogeneity among studies (I^2^ = 89%, *p* = 0.003) prevent firm conclusions. This emphasizes the need for future RCTs and indicates that TNF-α responses to exercise are highly variable, potentially influenced by factors such as baseline inflammatory status, health condition, or recovery dynamics.

A closer examination of individual results reveals significant discrepancies between studies. Sloan et al. (2018) [22] reported a greater reduction in TNF-α within the control group, whereas Borges et al. (2019) [12] observed no meaningful difference between groups. These divergent outcomes may be explained by differences in sample characteristics, intervention design, and timing of post-intervention measurements. 

Authors have shown that TNF-α levels may decrease as early as 10 min after acute aerobic exercise [42]. However, regular aerobic exercise has also been linked to metabolic adaptations in adipose tissue, including increased lipolytic capacity, enhanced fatty acid utilization, and mitochondrial biogenesis [43]. Although TNF-α has traditionally been associated with pro-inflammatory effects and metabolic dysfunction in obesity, its role may be more complex, particularly in the context of chronic exercise. It is important to consider that the presence of comorbid conditions may also influence circulating TNF-α levels. One proposed mechanism is that a transient elevation in TNF-α during exercise may regulate proteins involved in lipolysis, such as CIDEC [44]. CIDEC functions as a negative regulator of lipolysis in adipocytes by modulating adipose triglyceride lipase (ATGL) activity. It has been suggested that CIDEC directly interacts with ATGL, inhibiting its catalytic function in triglyceride breakdown [33].

However, current evidence is inconclusive. Further research is needed to explore the adaptive potential of TNF-α in response to exercise, particularly using well-controlled protocols in diverse populations. Longitudinal and mechanistic studies are needed to clarify whether transient TNF-α responses represent beneficial adaptations or context-dependent inflammatory signals.

### 4.6. Strengths

(I) This systematic review and meta-analysis performed a search across two institutional databases from the National Autonomous University of Mexico and the Autonomous University of Ciudad Juarez that includes PubMed, SCOPUS, Web of Science, EBSCOhost, and ScienceDirect. An additional search was conducted in PubMed. (II) The search strategy incorporates multiple synonyms and MeSH terms related to sleep quality, insomnia, sleep disturbances, IL-6, TNF-α, and exercise, thereby reducing the risk of missing relevant studies. (III) The risk of bias was independently assessed by two reviewers using the Cochrane Handbook for Systematic Reviews, thereby increasing the methodological transparency and reliability of the findings.

### 4.7. Limitations

A key limitation of this meta-analysis is the small number of studies that met the inclusion criteria. Considerable heterogeneity was observed across the studies, likely due to differences in participant characteristics (e.g., age, BMI, baseline inflammation, and the presence of sleep disorders), exercise protocols (e.g., duration, intensity, modality), and timing of biomarker assessment.

Another relevant limitation is the relatively short duration of most included interventions, which may be insufficient to reflect chronic adaptations to exercise, as shorter protocols (e.g., 16 weeks) tend to yield only subjective sleep improvements, while longer interventions (e.g., 12 months) have demonstrated significant effects on objective sleep outcomes [45]. The acute phase of physical training, particularly in sedentary individuals, is often accompanied by transient elevations in pro-inflammatory cytokines, which may temporarily worsen sleep quality [46]. This underscores the importance of long-term protocols and individualized exercise prescriptions to more accurately assess sustained immunological and sleep-related outcomes.

### 4.8. Publication of Bias

Visual inspection of funnel plots (Figure A1) revealed moderate asymmetry for sleep-related outcomes, particularly for PSQI (Panel A), suggesting the presence of small-study effects and potential publication bias. For ISI outcomes (Panel B), the funnel plot also exhibited visual asymmetry, although the small number of studies (*n* = 5) reduces the reliability of this assessment. Inflammatory markers (Panels C and D, representing IL-6 and TNF-α, respectively) included only two studies each, precluding any meaningful evaluation of symmetry or publication bias.

## 5. Conclusions

Aerobic exercise provides modest benefits for sleep quality and insomnia symptoms. Although the overall combined effect reached statistical significance, high heterogeneity among the studies weakened the strength of these findings. Additionally, the results related to inflammatory markers (IL-6 and TNF-α) are inconclusive and should be interpreted with caution, as only two trials were included, which limits the reliability of the estimates and their directionality; therefore, they should be regarded as exploratory findings. Consequently, more RCTs are necessary to confirm the potential of aerobic exercise as a strategy to treat insomnia, enhance sleep quality, and regulate immune–metabolic functions.

## Figures and Tables

**Figure 1 cimb-47-00572-f001:**
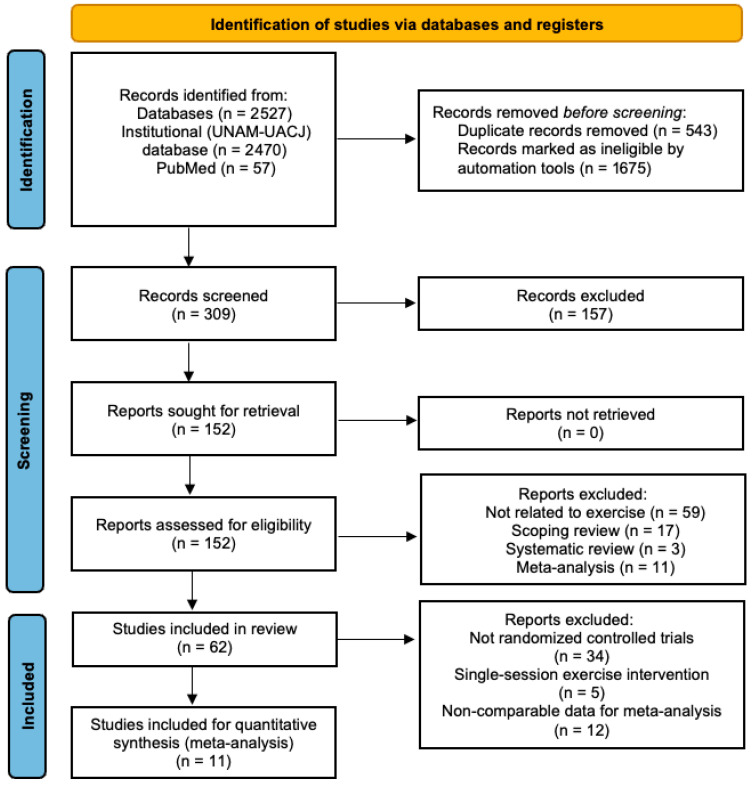
Flowchart of article selection.

**Figure 2 cimb-47-00572-f002:**
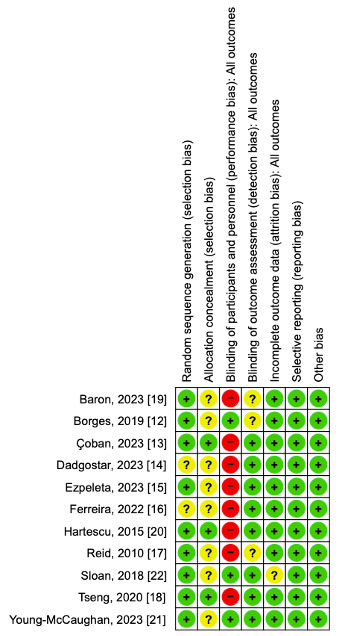
Risk of bias summary. Green, low risk of bias; yellow, unclear risk of bias; red, high risk of bias [12,13,14,15,16,17,18,19,20,21,22].

**Figure 3 cimb-47-00572-f003:**
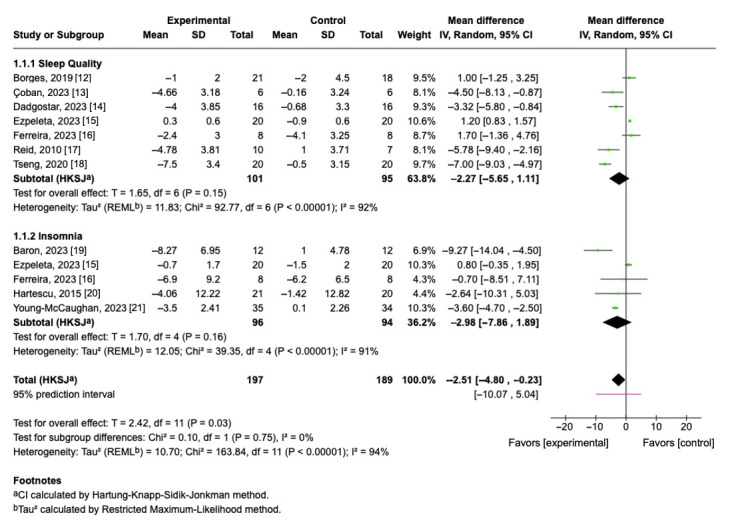
Forest plot of mean differences with 95% confidence intervals representing differences in sleep quality and insomnia in aerobic exercise group and controls [12,13,14,15,16,17,18,19,20,21]. Green squares indicate the mean differences of individual studies; horizontal lines represent the corresponding 95% confidence intervals. Black rhombuses indicate the pooled effect sizes with 95% confidence intervals.

**Figure 4 cimb-47-00572-f004:**
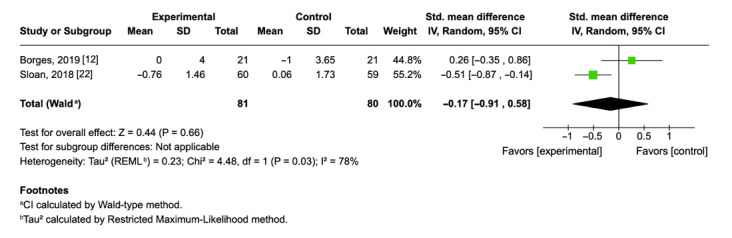
Forest plot of mean differences with 95% confidence intervals representing differences in circulating IL-6 in the aerobic exercise group and controls [12,22]. Green squares indicate the mean differences of individual studies; horizontal lines represent the corresponding 95% confidence intervals. Black rhombuses indicate the pooled effect sizes with 95% confidence intervals.

**Figure 5 cimb-47-00572-f005:**
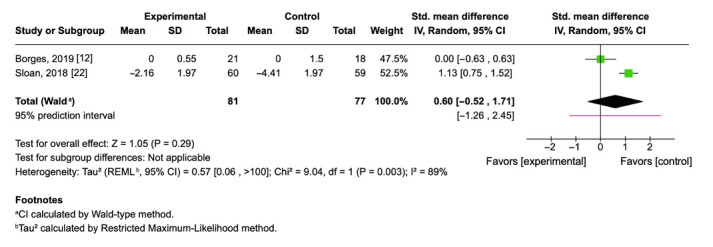
Forest plot of mean differences with 95% confidence intervals representing differences in circulating TNF-α in aerobic exercise group and controls [12,22]. Green squares indicate the mean differences of individual studies; horizontal lines represent the corresponding 95% confidence intervals. Black rhombuses indicate the pooled effect sizes with 95% confidence intervals.

**Table 1 cimb-47-00572-t001:** Descriptive characteristics of the included studies on sleep quality.

Study	Participants, Age (years)	Exercise Protocol	Δ Sleep Quality (PSQI)
Borges, 2019 [12]	22 males/17 females (50 ± 11.3)	Moderate aerobic exercise (treadmill walking); 3 times/week; ~50 min/session; 8 weeks	C: −0.44 ± 0.11 E: −0.5 ± 0.1
Çoban, 2023 [13]	6 males/6 females (44 ± 11.6)	Aerobic exercise on a treadmill; 3 times/week; 40 min/session; 8 weeks	C: −0.05 ± 0.33 E: −1.47 ± 0.42
Dadgostar, 2023 [14]	12 males/20 females (43.7 ± 0)	Combined exercise (aerobic and strength); 6 days/week (3 aerobic + 3 strength); 30–60 min/session; 12 weeks	C: −0.21 ± 0.13 E: −1.04± 0.14
Ezpeleta, 2023 [15]	8 males/32 females (44 ± 3)	Moderate aerobic exercise (treadmill, bike, or elliptical); 5 times/week; 60 min/session; 12 weeks	C: −1.5 ± 0.16 E: 0.5 ± 0.11
Ferreira, 2022 [16]	3 males/13 females (44.9 ± 9.58)	Moderate aerobic exercise (treadmill); 3 times/week; 50 min/session; 12 weeks	C: −1.26 ± 0.35 E: −0.8 ± 0.29
Reid, 2010 [17]	1 male/16 females (62.6 ± 4.35)	Moderate aerobic exercise (walking, bike, or treadmill); 4 times/week; 10–40 min/session; progressive intensity (55–75% HRmax); 16 weeks	C: 0.27 ± 0.29 E: −1.25 ± 0.28
Tseng, 2020 [18]	7 males/33 females (61.65 ± 7.10)	Moderate aerobic exercise (treadmill); 3 times/week; 50 min/session; 12 weeks	C: −0.16 ± 0.10 E: −2.21 ± 0.22

ΔPSQI values indicate the change in sleep quality, expressed as effect sizes (Cohen’s d) with their corresponding standard deviations, calculated as the difference between pre- and post-intervention scores within each group (C: control, E: exercise). Data used to estimate the magnitude and variability of the intervention effect for the meta-analysis.

**Table 2 cimb-47-00572-t002:** Descriptive characteristics of the included studies on insomnia.

Study	Participants, Age (years)	Exercise Protocol	Δ Insomnia (ISI)
Baron, 2023 [19]	24 females (45.7 ± 6.8)	Moderate–vigorous aerobic exercise (active walking outdoors or on a treadmill); 3 sessions/week; 75 min/session; 12 weeks	C: 0.21 ± 0.17 E: −1.19 ± 0.23
Ezpeleta, 2023 [15]	8 males/32 females (44 ± 3)	Moderate aerobic exercise (treadmill, bike, or elliptical); 5 times/week; 60 min/session; 12 weeks	C: −0.75 ± 0.11 E: −0.41 ± 0.10
Ferreira, 2022 [16]	3 males/13 females (44.9 ± 9.58)	Moderate aerobic exercise (treadmill); 3 times/week; 50 min/session; 12 weeks	C: −0.95 ± 0.31 E: −0.75 ± 0.29
Hartescu, 2015 [20]	10 males/30 females (59.8 ± 9.49)	Moderate-intensity brisk walking; ≥5 days/week; ≥30 min/day; 6 months	C: −0.11 ± 0.10 E: −0.33 ± 0.10
Young-McCaughan, 2023 [21]	63 males/5 females (35.5 ± 7.21)	Moderate aerobic exercise; 5 times/week; 20–25 min/session; 8 weeks	C: 0.04 ± 0.06 E: −1.45 ± 0.09

Δ ISI values indicate the change in insomnia severity, expressed as effect sizes (Cohen’s d) with their corresponding standard deviations, calculated from pre- and post-intervention scores within each group (C: control, E: exercise). Data used to evaluate the magnitude and variance of exercise on insomnia severity for the meta-analysis.

**Table 3 cimb-47-00572-t003:** Descriptive characteristics of the IL-6 studies included in this review.

Study	Participants, Age (years)	Exercise Protocol	Δ IL-6 (pg∙mL ^−1^)
Borges, 2020 [12]	22 males/17 females (50 ± 11.3)	Moderate aerobic exercise (treadmill walking); 3 times/week; ~50 min/session; 8 weeks	C: −0.27 ± 0.11 E: 0.0 ± 0.10
Sloan, 2018 [22]	56 males/63 females (31.3 ± 5.96)	Progressive aerobic exercise (cycling, treadmill, StairMaster); 4 times/week; 40–55 min/session; 12 weeks	C: −0.03 ± 0.03 E: −0.55 ± 0.04

Δ IL-6 values indicate the change in cytokine levels, expressed as effect sizes (Cohen’s d) with their corresponding standard deviations, calculated from pre- and post-intervention in both groups. These data supported the estimation of the exercise-induced effect on IL-6 concentrations.

**Table 4 cimb-47-00572-t004:** Descriptive characteristics of the TNF-α studies included in this review.

Study	Participants, Age (years)	Exercise Protocol	Δ TNF-α (pg∙mL ^−1^)
Borges, 2020 [12]	22 males/17 females (50 ± 11.3)	Moderate aerobic exercise (treadmill walking); 3 times/week; ~50 min/session; 8 weeks	C: 0.0 ± 0.1 E: 0.0 ± 0.1
Sloan, 2018 [22]	56 males/63 females (31.3 ± 5.96)	Progressive aerobic exercise (cycling, treadmill, StairMaster); 4 times/week; 40–55 min/session; 12 weeks	C: −2.22 ± 0.05 E: −0.62 ± 0.03

Δ TNF-α values indicate the change in cytokine levels, expressed as effect sizes (Cohen’s d) and their standard deviations, calculated from the differences between pre- and post-intervention measurements. Data used to determine the magnitude and variance of the intervention effect on TNF-α levels for inclusion in the meta-analysis.

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
