# Peer review of "Effects of Aerobic Exercise on Sleep Quality, Insomnia, and Inflammatory Markers: A Systematic Review and Meta-Analysis"

_cimb, 2025, doi:10.3390/cimb47070572_

Round 1

Reviewer 1 Report

Comments and Suggestions for Authors

Congratulations for this very important review in evaluating a problem with increasing incidence worldwide: sleep disorders and the possibility of eliminating them as naturally as possible. Regular physical exercise is followed by an improvement in sleep but always after a period of adaptation. Also, during the initial period of physical activity, sometimes for a period of several weeks, there is an increase in inflammation markers due to the body's adaptation processes to the new way of life, initially there may even be an increase in both inflammation and sleep disorders. Therefore, I believe that the limiting factor of this review is the need for long-term research on the interconnection of physical exercise adapted to the level of training - sleep disorders - inflammation.

I think this aspect should be emphasized.

Author Response

Coments:

Congratulations for this very important review in evaluating a problem with increasing incidence worldwide: sleep disorders and the possibility of eliminating them as naturally as possible. Regular physical exercise is followed by an improvement in sleep but always after a period of adaptation. Also, during the initial period of physical activity, sometimes for a period of several weeks, there is an increase in inflammation markers due to the body's adaptation processes to the new way of life, initially there may even be an increase in both inflammation and sleep disorders. Therefore, I believe that the limiting factor of this review is the need for long-term research on the interconnection of physical exercise adapted to the level of training - sleep disorders - inflammation.

I think this aspect should be emphasized.

Response: Thank you for this insightful comment. We have expanded the Limitations section to emphasize that most included interventions were of relatively short duration, which may be insufficient to capture long-term adaptations. We now reference the studies comparing 16-week vs. 12-month exercise programs, noting that only the longer interventions resulted in significant improvements in objective sleep outcomes. Additionally, we discuss the initial inflammatory response to physical activity in sedentary individuals as a possible explanation for transient worsening of sleep, reinforcing the need for individualized and longer-duration exercise protocols. These additions aim to address the complex interplay between exercise adaptation, sleep, and inflammation.

Reviewer 2 Report

Comments and Suggestions for Authors

This systematic review and meta-analysis addresses a timely and clinically relevant topic at the intersection of physical activity, sleep health, and systemic inflammation. The manuscript demonstrates scientific rigor in its adherence to PRISMA guidelines and its application of validated instruments. The authors provide a structured and comprehensive analysis of randomized controlled trials examining the effects of aerobic exercise on subjective sleep quality, insomnia symptoms, and inflammatory biomarkers. The effort to synthesize data from multiple institutional databases and to quantify effects via standardized meta-analytical methods is commendable. The search strategy is thorough and well-documented, including multiple academic databases and well-defined inclusion/exclusion criteria. The article follows a logical structure through methods, results, and interpretation. Also, the use of Cochrane RoB2 enhances the reliability of the included data.

I do have a series of observations that could enhance the overall quality and addressability of the manuscript.
Although the pooled effect of aerobic exercise on sleep parameters achieved statistical significance, subgroup analyses for PSQI and ISI were not individually significant. This discrepancy deserves a more nuanced discussion regarding clinical versus statistical significance and implications for different patient subgroups (individuals with versus without diagnosed insomnia).

Considering inflammatory markers, the analysis reveals non-significant effects of exercise on IL-6 and TNF-α levels, with substantial heterogeneity. The authors correctly discuss the dual role of IL-6, inflammatory and metabolic, and the complexity of TNF-α signaling. However, it would be valuable to more explicitly caution against overinterpreting the observed directions of change in cytokine levels given the limited number of included studies (n=2 for each marker).

The substantial heterogeneity across many analyses is a key limitation. While the authors identify several contributing factors as variation in protocol duration, intensity, population characteristics, there is little quantification or stratification. Meta-regression or further subgroup analysis (by BMI or baseline inflammation status) would have strengthened the findings.

Despite the title and objectives referencing both aerobic and anaerobic interventions, the meta-analysis focuses exclusively on aerobic protocols. The exclusion of anaerobic exercise, though justified due to insufficient data, should be reflected in the title and abstract.

The funnel plots suggest potential publication bias, but this is only briefly mentioned. A fuller discussion is warranted.

Minor language editing could enhance clarity in some sections (repeated phrases like "this study" or "this trial" could be varied stylistically).

Author Response

Comment 1

This systematic review and meta-analysis addresses a timely and clinically relevant topic at the intersection of physical activity, sleep health, and systemic inflammation. The manuscript demonstrates scientific rigor in its adherence to PRISMA guidelines and its application of validated instruments. The authors provide a structured and comprehensive analysis of randomized controlled trials examining the effects of aerobic exercise on subjective sleep quality, insomnia symptoms, and inflammatory biomarkers. The effort to synthesize data from multiple institutional databases and to quantify effects via standardized meta-analytical methods is commendable. The search strategy is thorough and well-documented, including multiple academic databases and well-defined inclusion/exclusion criteria. The article follows a logical structure through methods, results, and interpretation. Also, the use of Cochrane RoB2 enhances the reliability of the included data.

Response: We sincerely thank the reviewer for the positive and encouraging feedback. We appreciate your recognition of the methodological rigor, the clarity of the structure, and the relevance of the topic. No changes were made in response to this comment.

Comment 2

 I do have a series of observations that could enhance the overall quality and addressability of the manuscript.

Although the pooled effect of aerobic exercise on sleep parameters achieved statistical significance, subgroup analyses for PSQI and ISI were not individually significant. This discrepancy deserves a more nuanced discussion regarding clinical versus statistical significance and implications for different patient subgroups (individuals with versus without diagnosed insomnia).

Response: Thank you for this important observation. We have expanded the Discussion to include a more detailed interpretation of clinical versus statistical significance for the PSQI and ISI outcomes. Specifically, we now refer to published minimal important change (MIC) and minimal clinically important difference (MCID) values drawn from a recent publication, which report that the MIC for PSQI is approximately 3 points and that for ISI between 3 and 8 points, with frequently used MCID of 4 points. Our subgroup means changes (MD = -2.27 for PSQI and MD = -2.98 for ISI) did not meet these clinical thresholds, despite being in the expected direction. We also highlight those participants with diagnosed insomnia showed larger improvements than those without, reinforcing the importance of baseline symptom severity when interpreting subgroup results.

Comment 3

Considering inflammatory markers, the analysis reveals non-significant effects of exercise on IL-6 and TNF-α levels, with substantial heterogeneity. The authors correctly discuss the dual role of IL-6, inflammatory and metabolic, and the complexity of TNF-α signaling. However, it would be valuable to more explicitly caution against overinterpreting the observed directions of change in cytokine levels given the limited number of included studies (n=2 for each marker).

Response: We appreciate the reviewer’s recommendation. We have added an integrative statement in sections 4.4, 45, and the conclusion, emphasizing that both IL-6 and TNF-α results are based on only two eligible trials each. We now explicitly state that pooled estimates for these markers should be interpreted as exploratory and not conclusive, due to the limited data.

Comment 4

The substantial heterogeneity across many analyses is a key limitation. While the authors identify several contributing factors as variation in protocol duration, intensity, population characteristics, there is little quantification or stratification. Meta-regression or further subgroup analysis (by BMI or baseline inflammation status) would have strengthened the findings.

Response: Thank you for your comment. The impact of the few studies found, along with their high heterogeneity, is added to the discussion. We agree that a meta-regression including covariates such as nutritional and physical activity status, along with a sensitivity analysis, would strengthen the observed results; however, two studies remain insufficient. Additionally, we unfortunately did not have access to the databases of the reported studies.

Comment 5

Despite the title and objectives referencing both aerobic and anaerobic interventions, the meta-analysis focuses exclusively on aerobic protocols. The exclusion of anaerobic exercise, though justified due to insufficient data, should be reflected in the title and abstract.

Response: We appreciate this observation. The title has been revised to reflect the exclusive focus on aerobic exercise. In addition, the Abstract section now clarifies that anaerobic protocols were excluded.

Comment 6

The funnel plots suggest potential publication bias, but this is only briefly mentioned. A fuller discussion is warranted.

Response: Thank you for your valuable observation. We have added a dedicated subsection (4.8 Publication Bias), where we now provide a more detailed interpretation of funnel plots. 

Comment 7

Minor language editing could enhance clarity in some sections (repeated phrases like "this study" or "this trial" could be varied stylistically).

Response: We appreciate the reviewer’s suggestions regarding language clarity. We have carefully revised the manuscript to improve readability by reducing repetitive phrasing. Alternative terms and sentence structures have been incorporated where appropriate to enhance linguistic variety and flow, without compromising clarify and scientific accuracy.